# Factors Influencing False-Negative Results of QuantiFERON-TB Gold In-Tube (QFT-GIT) in Active Tuberculosis and the Desirability of Resetting Cutoffs for Different Populations: A Retrospective Study

**DOI:** 10.3390/tropicalmed7100278

**Published:** 2022-09-30

**Authors:** Yuanyuan Yu, Yidian Liu, Lan Yao, Yanheng Shen, Qin Sun, Wei Sha

**Affiliations:** 1Department of Tuberculosis, Shanghai Pulmonary Hospital, School of Medicine, Tongji University, Shanghai 200433, China; 2Clinical and Research Center for Tuberculosis, Shanghai Key Laboratory of Tuberculosis, Shanghai Pulmonary Hospital, School of Medicine, Tongji University, Shanghai 200433, China

**Keywords:** active tuberculosis, cutoff values, influencing factors, QFT-GIT

## Abstract

Objectives The value of QuantiFERON-TB Gold In-Tube (QFT-GIT) in the diagnosis of TB varies by population, comorbidities, and other factors. In this study, we aimed to investigate factors that influence false-negative results of QFT-GIT test in the diagnosis of TB as well as the impact of different cutoffs on the diagnostic value. Methods A total of 3562 patients who underwent QFT-GIT tests at Shanghai Pulmonary Hospital were enrolled retrospectively between May 2016 and May 2017. False-negative and false-positive results were analyzed using different clinical stratifications. The optimal cutoff values were established under different clinical conditions. Results Positive QFT-GIT results greatly shortened the time taken to diagnose smear-negative TB. The factors of age, smear and culture results, site of TB, comorbidity with tumors, white blood cell count, neutrophil count, and CD4/CD8 ratio were significantly correlated with false-negative QFT-GIT results (*p* < 0.05). Personalized cutoff values were established according to different influencing factors. The results showed high consistency between the smear-negative and total populations. Conclusion QFT-GIT can facilitate the early diagnosis of smear-negative TB. The diagnostic performance of the QFT-GIT test in the diagnosis of active TB was shown to be affected by many clinical factors. Personalized cutoff values may have superior value in the identification of active tuberculosis under different conditions.

## 1. Introduction

Tuberculosis (TB) is a communicable disease caused by *Mycobacterium tuberculosis* (MTB). The disease typically affects the lungs (Pulmonary TB, PTB) but can affect other sites (Extrapulmonary TB, EPTB). It has been estimated that approximately 10 million people fall ill with PTB each year, resulting in 1.4 million deaths every year worldwide. It is one of the top ten infectious fatal diseases. According to the Global TB Report in 2021 [1], China had the second-highest disease burden (8.5%) in 2020. Although the prevalence of TB is decreasing, its decline has been very slow in recent years. Therefore, early diagnosis and treatment are effective strategies for preventing the further spread of MTB infection. However, the clinical manifestations of TB are sometimes subtle, making early identification challenging, especially in the case of smear-negative TB.

The tuberculin skin test (TST) and the interferon-gamma release assay (IGRA) test are two currently available indirect diagnostic techniques used for the diagnosis of tuberculosis. The TST may cross-react with nontuberculous mycobacteria (NTM) or Bacille Calmette-Guérin (BCG), resulting in false-positive results, making it ineffective in the early diagnosis of TB. The IGRA includes the QuantiFERON-TB Gold in-tube (QFT-GIT) test and the T-SPOT.TB test [2]. The QFT-GIT test is the third generation of IGRA’s enzyme-linked immunosorbent assay (ELISA), used for the diagnosis of MTB infection as well as latent TB infection (LTBI). It uses the MTB-specific antigens ESAT-6, CFP-10, and TB7.7 to stimulate a patient’s immune lymphocyte T cells to produce γ-interferon (IFN-γ); then, the IFN-γ level is quantitatively measured using ELISA, and the results are determined according to the judgment standard [3,4]. QFT-GIT offers greater specificity than the TST [5,6,7].

In addition, once infected with LTBI, most people remain asymptomatic and are not contagious; however, 5–10% of cases may progress to active TB in the individual’s lifetime and become infectious [8]. In a study by Nijhawan et al. [9], it was reported that QFT-GIT test was more accurate compared to the TST in diagnosing latent MTB infection. According to a study by Amina Ahmed et al. [10], IGRAs are no less accurate in their predictive value than the TST, but they offer high specificity and negative predictive values in children <15 years of age. However, this test cannot be used to distinguish between active TB and LTBI. QFT-GIT may also be used as a potential tool for monitoring the effectiveness of antituberculosis treatment [11]. Chang et al. [12] found that 87.2% of patients displayed a significant decrease in IFN-γ levels detected via QFT-GIT after 2 months of treatment.

As QFT-GIT is an immunological detection technology, its sensitivity and specificity may be influenced by the host’s immune condition. In Bae W, P.K.’s research, it was found that the diagnostic sensitivity of both QFT-GIT and T-SPOT declined with age, and the effect on QFT-GIT was more substantial [13]. In addition, the factors of sex, immunodeficiency (such as glucocorticoid therapy, HIV-infected immunosuppression, or tumor necrosis factor antagonist therapy), different sites of MTB infection, smoking, diabetes, etc., may have effects on false-negative or uncertain QFT-GIT results [11]. In a study by Edwards et al. [14], dexamethasone converted QFT-GIT results from positive to negative in 30% of participants; antigen-stimulated IFN-γ, IL-2 and TNF-α responses were markedly reduced; and infliximab caused QFT-GIT results conversion in up to 30% of participants and substantial reductions in all cytokine responses, indicating that corticosteroids and infliximab impair the performance of IFN-γ release assays used for the diagnosis of latent tuberculosis. Magee et al. [15] indicated that a high IFN-γ TB antigen response was more common among those with pre-diabetes (aOR 1.9, 95% CI 1.0–3.6) than euglycemic participants. Kim et al. [16] found that the levels of false-negative QFT-GIT results were different depending on the site of infection. In the proven TB group, negative QFT-GIT results comprised 28.6% (95% CI 0.04–0.71) of pleural TB cases, 8.3% (95% CI 0.002–0.38) of lymph node TB cases, 8.3% (95% CI 0.002–0.38) of skeletal TB cases, and 5.8% (95% CI 0.001–0.28) of gastrointestinal TB cases. Among probable TB cases, negative QFT-GIT results were identified in 46.2% (95% CI 0.19–0.75) of skeletal TB cases, 33.3% (95% CI 10–0.65) of pericardial TB cases, 30.8% (95% CI 0.09–0.61) of pleural TB cases, and 17.2% (95% CI 0.10–0.56) of gastrointestinal TB cases. Moreover, CD4 T-cell counts in patients have also been shown to affect the diagnosis of TB via QFT-GIT [17,18,19].

Very few articles have considered the influence of tumors on false-negative QFT-GIT results, as well as white blood cells, neutrophils, etc. Furthermore, most of the previous studies in the literature only analyzed one influencing factor individually; these influencing factors were rarely analyzed together. The objectives of this study were to use statistical methods to comprehensively estimate the false-negative (as well as false-positive) QFT-GIT results in the Chinese population, to investigate the elements that impact the diagnostic value and to recommend appropriate cutoff values for different national situations in China.

## 2. Materials and Methods

### 2.1. Study Populations

QFT-GIT was performed when a patient was suspected of having TB. In this study, the computerized records of 3725 patients who underwent QFT-GIT tests at Shanghai Pulmonary Hospital from May 2016 to May 2017 were retrospectively analyzed. After removing undiagnosed cases (18), those with uncertain QFT-GIT results (51), and those with old pulmonary TB (94), a total of 3562 patients were enrolled in the study (Figure 1), including 2251 males and 1311 females, aged 11–97 years. The study was conducted according to the guidelines of the Declaration of Helsinki, and approved by the Ethics Committee of Shanghai Pulmonary Hospital (K18-215Z, January 2019 and 31 December 2021). All the patients completed informed consent forms before undergoing the QFT-GIT test, because the cost of this test was not covered by medical insurance.

### 2.2. Patient Classification and Diagnostic Criteria

Patients were categorized as active TB or non-TB individuals based on data from both inpatient and outpatient electronic records. Those with active TB included individuals with PTB (confirmed PTB and clinically diagnosed PTB) and EPTB. The diagnostic criteria were based on Chinese guidelines [20]. The specific criteria are as follows:

Confirmed PTB: (1) Chest imaging showed lesions consistent with active PTB, with positive sputum/bronchoalveolar lavage fluid (BALF) smear and/or culture that were identified as MTB. (2) Chest imaging showed lesions consistent with active PTB, and the pulmonary biopsy specimen was pathologically diagnosed as tuberculosis.

Clinically diagnosed PTB: (1) Chest imaging showed lesions suspected to be PTB; (2) the sputum and/or BALF smear and culture results were negative; (3) TST or IGRA results were positive; (4) other pulmonary diseases were excluded; or (5) the individual had a therapeutic response after 2 months of anti-tuberculosis treatment.

EPTB: (1) Extra-pulmonary biopsy was bacteriologically or pathologically confirmed as tuberculosis. (2) EPTB was suspected from clinical symptoms and radiological findings, positive TST or IGRA results were shown, other diseases were excluded, and patients had a therapeutic response after anti-tuberculosis treatment.

Non-TB: The clinical or etiological exclusion of TB or culture confirmed as NTM.

### 2.3. Data Collection

Each patient’s age (<18 years (n = 65), 18–40 years (n = 1102), 40–60 years (n = 1123), or >60 years (n = 1272)); sex (male (n = 2251) or female (n = 1311)); smear results (positive (n = 748) or negative (n = 1400)); culture results (positive (n = 1524) or negative (n = 706)) and drug sensitivity test (DST) results (sensitive (n = 704), DR-TB (n = 653) or RR-TB (n = 178)); TB treatment history (newly diagnosed TB (n = 1881) or retreatment TB (n = 349)); site of TB (intrapulmonary TB (n = 1937), EPTB (n = 67), or co-infection (n = 226); tuberculous pleuritis was included as intrapulmonary TB in this study); comorbidities (tumor (n = 251), diabetes (n = 426), occupational disease (n = 37), or immune system disease (n = 73)); the time of QFT-GIT results reported and time of smear-negative TB diagnosis; routine blood tests results (counts of white blood cells, neutrophils, and lymphocytes); and CD4 levels, CD8 levels, and CD4/CD8 ratios were collected from the electronic medical records.

### 2.4. QFT-GIT

QFT-GIT was usually performed on the second day of the first admission. Peripheral blood (4–6 mL) was collected from the patients in vacuum heparin anti-coagulation tubes; stored at room temperature; sent to the laboratory; packed into Nil tubes, TB Ag tubes and Mitogen tubes; and shaken immediately. After 16–24 h of incubation at 37 °C, the IFN-γ concentration was measured using an ELISA kit (Cellestis/QIAGEN, Melbourne, Australia). The test was performed, and the results were read according to the manufacturer’s guidelines. The Nil tubes acted as blank controls, the Mitogen tubes were mitogen-positive controls, and the results were based on IFN-γ concentrations in sample tubes minus the negative control values (IU/mL). If the IFN-γ concentration in the negative control tube was ≤8.0 IU/mL and the IFN-γ concentration in the sample tube minus that in the negative control was ≥0.35 and ≥25% of the negative control value, the result was positive. If the IFN-γ concentration in the negative control tube was ≤8.0 IU/mL, the positive control value minus the negative control value was ≥0.5, and the IFN-γ concentration in the sample tube minus that in the negative control was <0.35 or <25% of the negative control value, the result was negative. Other results were categorized as uncertain [11].

### 2.5. Statistical Analysis

The data were processed using SPSS version 23.0 software (IBM SPSS^®^ Statistics, Chicago, IL, USA). Count data were presented as the frequency and composition ratio. The time taken to diagnose smear-negative TB was analyzed with a nonparametric test. The true/false-positive and true/false-negative results, positive likelihood ratio (PLR), positive predictive value (PPV), negative likelihood ratio (NLR), and negative predictive value (NPV) were calculated using cases of active TB disease (defined as described above) as a reference. Logistic regression and chi-square tests were utilized to analyze the classification variables. A *p* value of <0.05 was regarded as statistically significant. To determine the comprehensive effect of influencing factors on true/false-negative and true/false-positive QFT-GIT results, multivariate logistic regression analysis was carried out using significant variables (*p* < 0.05) in univariate analysis.

## 3. Results

### 3.1. Study Population

In this retrospective analysis, a total of 3562 patients were included. The median age of the patients was 53 (interquartile range: 32–63), with 65 patients under 18 (1.8%), 2225 patients between 18 and 60 (including 18) (62.5%), and 1272 patients over 60 (including 60) (35.7%). There were 2251 males (63.2%) and 1311 females (36.8%) in total. A total of 2230 of the 3562 patients were eventually diagnosed with TB. The sensitivity and specificity of QFT-GIT in the diagnosis of active TB were 87.98% (1962/2230) and 71.17% (948/1332), respectively. In the NTM population, 27.06% (46/170) were positive according to the QFT-GIT, while the percentage was 29.09% (338/1162) in the non-TB and non-NTM populations, indicating no significant difference.

### 3.2. Positive QFT-GIT Results Greatly Shortened the Time Taken to Diagnose Smear-Negative TB

We took the QFT-GIT reporting time as the starting point and analyzed the time needed for TB diagnosis. To more effectively investigate the diagnostic value of QFT-GIT, in this stage of the study, we focused on smear-negative and newly diagnosed TB patients, and we excluded patients who were diagnosed with TB before QFT results were available. As shown in Table 1, the diagnosis time taken for QFT-GIT-positive patients was significantly shorter than the time taken for those with negative results.

### 3.3. Factors Influencing the False-Negative and True-Positive QFT-GIT Results in the Diagnosis of Active Tuberculosis

The variables that were included in the univariate analysis are shown in Table 2 and Appendix A. In the univariate analysis, the factors of age, smear and culture findings, site of TB, comorbidity with tumor, white blood cell count, neutrophil count, and CD4/CD8 ratios were substantially linked to true-positive (also known as sensitivity) and false-negative QFT-GIT outcomes. With increasing age, there was a statistically significant increase in false-negative outcomes. Both PLR and PPV decreased with aging (shown in Table 2). There were fewer false-negative QFT-GIT results in the intrapulmonary TB and intrapulmonary TB co-infection with EPTB groups than in the EPTB only group (12% vs. 19.4%, *p* = 0.068; 10.2% vs. 19.4%, *p* = 0.043) (chi-square analysis). The number of false-negative results was significantly higher in those with higher white blood cell or neutrophil counts than in the normal group (20.1% vs. 11.1%, *p* = 0.001; 21.4% vs. 11.5%, *p* = 0.002), while there was no statistically significant difference between those with lower white blood cell or neutrophil counts and the normal group. The PLR in the three white blood count groups was similar (approximately 3.0), but the PPV in the group with higher levels was lower (69% compared with 86% and 83%). The group with increased neutrophil count displayed the highest PLR (3.73), while the normal group had the highest PPV (85%). The number of false-negative results increased when the CD4/CD8 ratio fell (16.8% vs. 10.2%, *p* = 0.011). The PLR was greater in the decreased group, while the PPV was similar to that in the group with normal CD4/CD8 ratio but higher than that in the group with increased CD4/CD8 ratio.

Multivariate analysis was used to evaluate eight parameters that were significantly associated with true-positive and false-negative QFT-GIT outcomes. Only the factors of age, culture results, neutrophil counts, and CD4/CD8 ratio were significantly related to false-negative QFT-GIT outcomes, as shown in Table 3.

### 3.4. Factors Influencing the False-Positive and True-Negative QFT-GIT Results in the Diagnosis of Active Tuberculosis

We also carried out an analysis of factors that affected the false-positive and true-negative QFT-GIT results (Table 4 and Appendix A). The factors of age, sex, diabetes, immune system diseases, neutrophil counts, lymphocyte counts, and CD4/CD8 ratio were significantly correlated with false-positive and true-negative (known as specificity) QFT-GIT results. When these seven factors were analyzed using multivariate risk analysis, the results showed that age, sex, immune system diseases, and neutrophil counts still had a significant influence on false-positive QFT-GIT results (Table 5). The NPV was 100% in patients with occupational illness or increased lymphocyte counts, demonstrating that negative QFT-GIT results can practically rule out TB. The values in the groups with tumors, decreased neutrophil counts, and immune system diseases followed closely behind, with NPVs of 90.4%, 89.9%, and 89.5%, respectively.

### 3.5. Influence of Different Cutoff Values of the QFT-GIT on the Diagnostic Value

We identified the best critical values of separate factors for individual diagnoses. The optimal critical value of QFT-GIT in this study calculated using the receiver operating characteristic (ROC) curve was 0.355 (sensitivity of 84.7%, and specificity of 83.0%), and the corresponding critical values of seven influencing factors significantly related to QFT-GIT sensitivity and specificity were calculated. The results are presented in Table 6 (QFT-GIT > 10 does not show the specific values in SPSS software, and the data with value > 10 were recorded as 10 in the statistical analysis).

We also selected several influencing factors to confirm the diagnostic value of the QFT-GIT after adjusting the cutoff values. These factors were selected because they had sensitivity or specificity levels that were considerably below average. Additionally, some factors were excluded due to the small sample size. The results are shown in Table 7. After adjusting the cutoff values, analyses in the elder group (≥60 years) and the diabetes group had higher specificities but lower sensitivities. The sensitivity of the analysis in the tumor group improved after adjustment, but the specificity decreased. Therefore, regarding the optimal cutoff value, specific conditions should be specified in the diagnosis or exclusion of TB. Despite the fact that 0.35 was chosen as the universal cutoff value offering the best Youden index for all influencing groups, understanding the ideal cutoff value may be of better prospective value in diagnosing or excluding TB in different conditions.

### 3.6. Factors Influencing the False-Negative and False-Positive QFT-GIT Results in Smear-Negative Population

Furthermore, we analyzed the smear-negative population regarding false-negative and false-positive results and found that age, culture results, site of TB, white blood cell counts, neutrophil counts, lymphocyte counts, and CD8 were significantly associated with false-negative results (shown in Appendix A). A decreased CD4/CD8 ratio (*p* = 0.051) and comorbidity with a tumor (*p* = 0.057) were not statistically linked, although the *p* values were extremely close to 0.05. False-positive findings were strongly linked with age, sex, comorbidity with diabetes or immune system diseases, neutrophil counts, lymphocyte counts, CD4 levels, and CD4/CD8 ratio (shown in Appendix A). The effect of influencing factors on the diagnostic value of QFT in the smear-negative population was consistent with that in the entire population.

## 4. Discussion

QFT-GIT is widely used as a tool for the diagnosis of TB infection as well as latent TB infection (LTBI) and may also be used as a potential tool for monitoring the effectiveness of anti-tuberculosis treatment. However, QFT-GIT cannot be used to distinguish between active TB and LTBI, and as an immunological detection technology, its sensitivity and specificity may be influenced by the host’s immune condition. To fully understand the risk factors that influence false-negative/positive QFT-GIT results, we conducted this large-sample retrospective study, which included single-factorial as well as multifactorial analysis.

Positive QFT-GIT results greatly shortened the time taken to diagnose smear-negative TB; the median diagnosis time was reduced from 5 days to 2 days (*p* < 0.001), which confirmed the important value of QFT-GIT in the diagnosis of active tuberculosis. QFT-GIT facilitates the early diagnosis and prompt treatment of TB and prevents the spread of TB. Furthermore, QFT-GIT has irreplaceable advantages: it is quick and noninvasive. Patients from whom alveolar lavage fluid and tissue specimens cannot be obtained through invasive examination or from whom effective sputum samples cannot be obtained, such as the aged and critically ill, may benefit from QFT-GIT in early diagnosis. Thus, this means it is meaningful to investigate the risk factors affecting false-negative/positive results of the QFT-GIT.

The research results from the smear-negative population and the total population showed high consistency. However, there were exceptions to this. For example, comorbidity with a tumor was an independent factor related to false-negative results in the total population (*p* = 0.008) but not in the smear-negative population (*p* = 0.057), probably due to the smaller sample size. The PPV in tumor patients was only 44%, while the NPV was 90%, indicating that a negative result has a very good exclusion diagnosis value, but a positive result may lead to a misdiagnosis of TB. Patients with tumors and TB must initially receive anti-tuberculosis treatment as chemotherapy can induce severe tuberculosis recurrence, and misdiagnosis increases patients’ illness burden and delays tumor treatment, while a missed diagnosis of TB may be fatal for tumor patients. This means it is meaningful to adjust the cutoff value in a specific population.

Most of our results were consistent with previous studies or can be interpreted using related references. In this study, the sensitivity and specificity of the QFT-GIT were found to be 87.98% and 71.17%, respectively, which were in accordance with previous reports (approximately 84.2% and 74.5%, respectively) [21,22]. With increasing age, there was a statistically significant increase in false-negative outcomes, because children’s reaction rates to TB antigens were much higher than those in adults [3,23,24], and the IFN-γ response stimulated by TB antigens decreased dramatically with aging. The prevalence rate of TB in males (66.3%, 1492/2251) was significantly higher than that in females (56.3%, 738/1311) (χ^2^ = 35.309, *p* < 0.001), which was in agreement with most research reports [25], but according to Ting WY et al., the male sex is not a factor with substantial impact after adjusting for age, sex, smoking history, and other clinical characteristics [26]. This result may explain why in the multivariate risk analysis, there was no significant difference between males and females, which also suggested that sex is only a complicated reflection of multiple factors. In the analysis of the site of TB, we found that intrapulmonary co-infection with EPTB was an independent influencing factor for false-negative QFT-GIT results, which was similar to the results of the study by Kim et al. [16], who found that the false-negative rate varied depending on the site of TB infection, with the rate in patients with TB meningitis being the highest. Regarding tumor patients, tumor cells can reduce the number of effector T cells and limit their function and immunological activity [27], reducing their response to TB antigen activation, thus causing false-negative results, as concluded in this study.

Some results in this study differed from previous results or could not be explained by previous studies. In this study, the specificity of the QFT-GIT in diabetes mellitus (DM) patients was significantly lower (61.5% vs. 72.4%, *p* = 0.006), while the sensitivity showed no significant difference (90.6% vs. 87.6%, *p* = 0.141). Patients with diabetes or pre-diabetes were more likely to have a high IFN-γ TB antigen response than euglycemic participants (OR 1.9, 95% CI: 1.0–3.6) [15]. However, another study on patients with latent TB infection showed that the response to MTB antigens was lower in patients with DM than in patients without DM [28]. DM can lead to immune system disorder, causing cytokine and chemokine levels to fluctuate [15,28,29]. The incidence of pneumoconiosis complicated by TB in China is 14.8% [30], and it was found that the spontaneous release of IFN-γ in the alveolar lavage fluid of patients with asbestosis was greater than that in healthy people [31,32], and this release may be affected by the regulation of regulatory T cells such as CD4^+^ and CD8^+^ T-cells (mostly driven by CD4^+^ T-cell reactions), which were disturbed by silica [33,34,35,36]. However, in this study, there was no significant difference between analyses performed in patients with occupational diseases and the normal population in terms of either sensitivity or specificity, possibly due to the small sample size. When patients had immune system diseases, the sensitivity and specificity level both increased, but there was only a significant difference in specificity (*p* = 0.01). Regarding patients who have immune system diseases, treatments with steroid hormones, biological response regulators, and antitumor necrosis factor-α (TNF-α) are closely linked to the reactivation of TB [37]. A deficiency in the costimulatory function of antigen-presenting cells could be one reason for the reduced cellular immunity [38].

Regarding blood cells, in this study, increased white blood cell and neutrophil counts, decreased lymphocyte counts, and CD4/CD8 ratios were risk factors influencing the value of the QFT-GIT results. Peripheral blood lymphocytes are the main immune cells that release interferon; therefore, interferon release varies with lymphocyte activity, which was confirmed in the research by Komiya et al. [39]. The neutrophil/lymphocyte ratio was found to be an independent predictor of uncertain QFT-GIT results [40] and was significantly lower after treatment than that in untreated patients [41]. Due to the complex immune response of the body, it was difficult to determine a clear relationship between white blood cells, neutrophil cells and interferon release to explain these results. This requires further study.

## 5. Conclusions

To summarize, QFT-GIT can help facilitate the early diagnosis of smear-negative TB. The diagnostic performance of the QFT-GIT in the diagnosis of active TB is affected by many factors, and tailored judgments based on the patient’s individual situation may enable a more accurate TB diagnosis. In the future, personalized cutoff values may be more useful in detecting active tuberculosis in different situations.

There are certain limitations to our research. Due to the retrospective properties of this study, various medical/case histories, glycemic control status, BMI, and the treatment of complications, etc., were not systematically documented. Many groups were not subdivided, such as different types and stages of tumors. The sample data in some groups were too limited to be analyzed successfully. Additionally, a study regarding LTBI populations needs to be conducted in the future.

## Figures and Tables

**Figure 1 tropicalmed-07-00278-f001:**
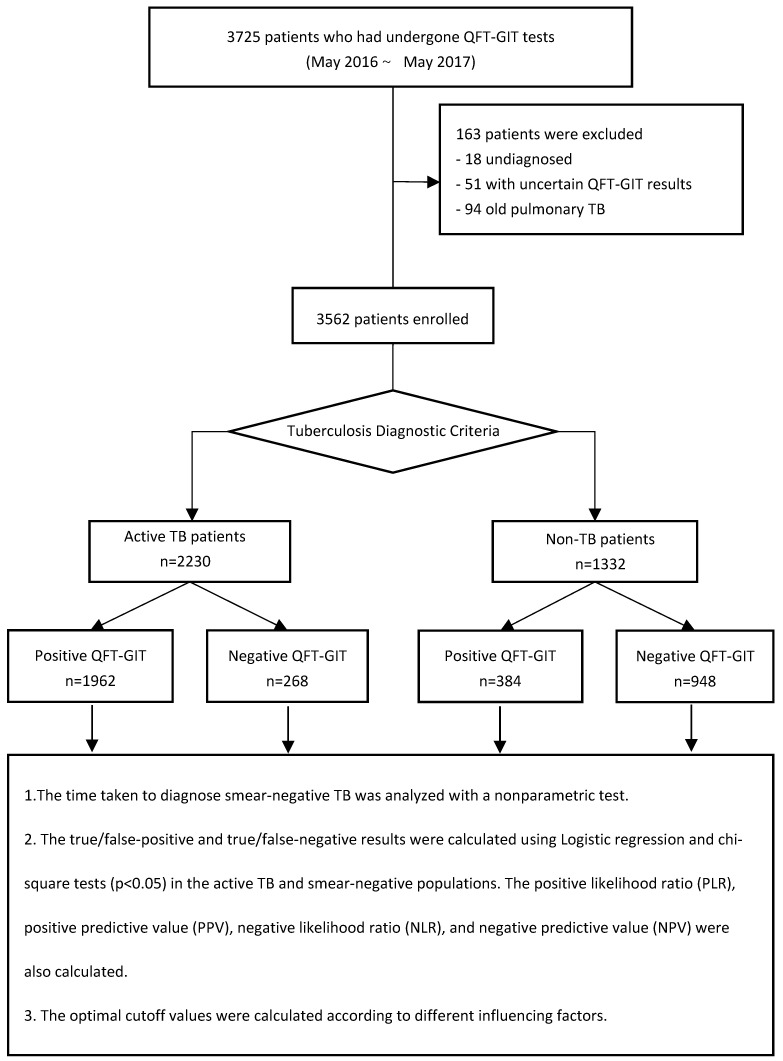
Initially, 3725 patients who underwent QFT-GIT tests from May 2016 to May 2017 were retrospectively analyzed. After excluding 163 cases, a total of 3562 patients were enrolled in the study. n, number.

**Table 1 tropicalmed-07-00278-t001:** Positive QFT-GIT results greatly shortened the time to diagnose TB. n: number. d: diagnosis time.

	Median Diagnosis Time (d)	Average Diagnosis Time (d)	Mode (d)	*p*
QFT-GIT positive (n = 324)	2	5.15	0	*p* < 0.001
QFT-GIT negative (n = 98)	5	30.92	5	

**Table 2 tropicalmed-07-00278-t002:** Single-factor risk analysis of false-negative and true-positive QFT-GIT results in the diagnosis of active tuberculosis.

Influencing Factor	Group	True Positive n (%)	False Negative n (%)	OR (95% CI)	*p*	PLR ^1^	PPV ^2^ (%)
Age	<18 years	58 (98.3)	1 (1.7)	12.619 (1.728–92.144)	0.012	/	100
	18–40 years	869 (92.1)	75 (7.9)	2.521 (1.836–3.462)	<0.001	4.04	96.0
	40–60 years	557 (86.4)	88 (13.6)	1.377 (1.011–1.876)	0.042	3.30	81.7
	≥60 years	478 (82.1)	104 (17.9)	Contrast		2.54	68.2
Sex	Male	1310 (87.8)	182 (12.2)	0.949 (0.722–1.248)	0.709	2.62	83.4
	Female	652 (88.3)	86 (11.7)			3.89	83.4
Smear	Positive	707 (91.1)	69 (8.9)	0.615 (0.461–0.822)	0.001		
	Negative	1255 (86.3)	199 (16.7)				
Culture	Positive	1400 (91.9)	124 (8.1)	0.346 (0.267–0.448)	<0.001		
	Negative	562 (79.6)	144 (20.4)				
Treatment type	New	1663 (88.4)	218 (11.6)	1.276 (0.916–1.766)	0.149		
	Retreatment	299 (85.7)	50 (14.3)				
Site of TB	Intrapulmonary	1705 (88)	232 (12)	0.833 (0.530–1.309)	0.428		
	EPTB	54 (80.6)	13 (19.4)	0.471 (0.224–0.990)	0.047		
	Co-infection ^3^	203 (89.8)	23 (10.2)	Contrast			
DST	Sensitive	642 (91.2)	62 (8.8)	1.237 (0.719–2.129)	0.442		
	DR-TB ^4^	609 (93.3)	44 (6.7)	1.654 (0.939–2.912)	0.081		
	RR-TB ^5^	159 (89.3)	19 (10.7)	Contrast			
Tumor	Yes	46 (76.7)	14 (23.3)	2.296 (1.244–4.235)	0.008	2.48	43.8
	No	1916 (88.3)	254 (11.7)			3.10	85.5
Diabetes	Yes	261 (90.6)	27 (9.4)	0.730 (0.480–1.110)	0.141	2.35	82.1
	No	1701 (87.6)	241 (12.4)			3.17	83.9
Occupational disease	Yes	14 (100)	0 (0)	0	0.999	2.88	63.6
	No	1948 (87.9)	268 (12.1)			3.06	83.8
Immune system diseases ^6^	Yes	32 (88.9)	4 (11.1)	0.914(0.321–2.604)	0.866	10.96	91.4
	No	1930 (88.0)	264 (12.0)			2.99	83.5
WBC ^7^	Normal	1657 (88.9)	207 (11.1)	Contrast		2.98	85.5
	Decreased	126 (86.3)	20 (13.7)	1.271 (0.776–2.801)	0.341	3.21	83.4
	Increased	110 (79.7)	28 (20.3)	2.038 (1.313–3.162)	0.001	3.04	69.2
NE ^8^	Normal	1710 (88.5)	223 (11.5)	Contrast		2.90	85.3
	Decreased	91 (92.9)	7 (7.1)	0.590 (0.270–1.288)	0.185	3.33	79.1
	Increased	92 (78.6)	25 (21.4)	2.084 (1.311–3.313)	0.002	3.73	72.4
LC ^9^	Normal	1714 (88.5)	223 (11.5)	Contrast		2.98	83.6
	Decreased	175 (84.5)	32 (15.5)	1.405 (0.940–2.101)	0.097	4.71	92.6
	Increased	4 (100)	0 (0)	.	SE ^10^ = 0	2.67	57.1
CD4 ^11^	Normal	596 (88.3)	79 (11.7)	Contrast		2.71	82.7
	Decreased	316 (87.5)	45 (12.5)	1.074 (0.727–1.588)	0.719	3.36	84.7
	Increased	686 (88.4)	90 (11.6)	0.990 (0.718–1.365)	0.950	3.06	84.1
CD8 ^12^	Normal	557 (90.4)	59 (9.6)	Contrast		2.71	82.9
	Decreased	782 (87.3)	114 (12.7)	1.376 (0.987–1.919)	0.060	3.06	83.3
	Increased	259 (86.3)	41 (13.7)	1.494 (0.977–2.286)	0.064	3.41	86.6
CD4/CD8 ^13^ ratio	Normal	840 (89.8)	95 (10.2)	Contrast		4.33	89.8
	Decreased	149 (83.2)	30 (16.8)	1.780 (1.140–2.781)	0.011	4.50	89.8
	Increased	609 (87.2)	89 (12.8)	1.292 (0.950–1.757)	0.102	2.92	82.5
LC decline ratio	≥50%	15 (83.3)	3 (16.7)	0.877 (0.233–3.309)	0.847	5.83	93.8
	25–50%	46 (83.6)	9 (16.4)	0.897 (0.380–2.115)	0.803	5.85	93.9
	0–25%	114 (85.1)	20 (14.9)	Contrast		4.63	92.7

^1^ PLR: Positive likelihood ratio. ^2^ PPV: Positive predictive value. ^3^ Co-infection: Intrapulmonary co-infection with EPTB. ^4^ DR-TB: Drug-resistant TB while Rifampicin is sensitive. ^5^ RR-TB: Drug-resistant TB while Rifampicin is resistant. ^6^ Immune system disease: Including rheumatoid arthritis, ankylosing spondylitis, systemic lupus erythematosus, colitis, etc. ^7^ Regular reference value of WBCs (white blood cells): 4.00–10.00 × 10^9^/L. ^8^ Regular reference value of NEs (neutrophils): 2.00–7.70 × 10^9^/L. ^9^ Regular reference value of LCs (lymphocytes): 0.80–4.00 × 10^9^/L. ^10^ SE: Standard error. ^11^ Regular reference value of CD4: 35.5 ± 5.3%. ^12^ Regular reference value of CD8: 25.1 ± 4.3%. ^13^ Regular reference value of CD4/CD8 ratio: 1.54 ± 0.59.

**Table 3 tropicalmed-07-00278-t003:** Multivariate risk analysis of false-negative and true-positive QFT-GIT results in the diagnosis of active tuberculosis.

Influencing Factor	Group	True Positive n (%)	False Negative n (%)	OR (95% CI)	*p*
Age	<18 years	54 (98.2)	1 (1.8)	0.107 (0.014–0.813)	0.031
	18–40 years	844 (92.0)	73 (8.0)	0.398 (0.271–0.586)	<0.001
	40–60 years	539 (86.5)	84 (13.5)	0.753 (0.524–1.083)	0.126
	≥60 years	456 (82.5)	97 (17.5)	Contrast	
Sputum smear	Positive	681 (91.0)	67 (9.0)	1.219 (0.836–1.778)	0.304
	Negative	1212 (86.6)	188 (13.4)		
Sputum culture	Positive	1352 (92.2)	114 (7.8)	3.844 (2.745–5.384)	<0.001
	Negative	541 (79.3)	141 (20.7)		
Site of TB	Intrapulmonary	1646 (88.1)	223 (11.9)	1.516 (0.870–2.641)	0.142
	EPTB	53 (81.5)	12 (18.5)	1.487 (0.615–3.595)	0.378
	Co-infection	194 (90.7)	20 (9.3)	Contrast	
Tumor	Yes	43 (78.2)	12 (21.8)	0.691 (0.318–1.504)	0.352
	No	1850 (88.4)	243 (11.6)		
WBC	Normal	1657 (88.9)	207 (11.1)	Contrast	
	Decreased	126 (86.3)	20 (13.7)	1.716 (0.906–3.252)	0.098
	Increased	110 (79.7)	28 (20.3)	1.361 (0.530–3.494)	0.522
NE	Normal	1710 (88.5)	223 (11.5)	Contrast	
	Decreased	91 (92.9)	7 (7.1)	0.438 (0.160–1.200)	0.108
	Increased	92 (78.6)	25 (21.4)	0.132 (0.033–0.529)	0.004
CD4/CD8 ratio	Normal	840 (89.8)	95 (10.2)	Contrast	
	Decreased	149 (83.2)	30 (16.8)	0.950 (0.685–1.319)	0.761
	Increased	609 (87.2)	89 (12.8)	1.728 (1.066–2.802)	0.027

**Table 4 tropicalmed-07-00278-t004:** Single-factor risk analysis of false-positive and true-negative QFT-GIT results in the diagnosis of active tuberculosis.

Influencing Factor	Group	True Negative n (%)	False Positive n (%)	OR (95% CI)	*p*	NLR ^1^	NPV ^2^ (%)
Age	<18 years	6 (100)	0 (0)		SE = 0	59.00	85.7
	18–40 years	122 (77.2)	36 (22.8)	1.618 (1.080–2.425)	0.020	9.72	61.9
	40–60 years	353 (73.8)	125 (26.2)	1.349 (1.041–1.747)	0.024	5.41	80.0
	≥60 years	467 (67.7)	223 (32.3)	Contrast		3.79	81.8
Sex	Male	505 (66.5)	254 (33.5)	0.583 (0.456–0.747)	<0.001	5.45	73.5
	Female	443 (77.3)	130 (22.4)			6.63	83.7
Tumor	Yes	132 (69.1)	59 (30.9)	1.122 (0.805–1.565)	0.497	2.96	90.4
	No	816 (71.5)	325 (28.5)			6.11	76.3
Diabetes	Yes	91 (61.5)	57 (38.5)	1.642 (1.151–2.341)	0.006	6.56	77.1
	No	857 (72.4)	327 (27.6)			5.83	78.1
Occupational disease	Yes	15 (65.2)	8 (34.8)	1.323 (0.556–3.147)	0.526	/	100
	No	933 (71.3)	376 (28.7)			5.89	77
Immune system diseases	Yes	34 (91.9)	3 (8.1)	0.212 (0.065–0.693)	0.01	8.27	89.5
	No	914 (70.6)	381 (29.4)			5.87	77.6
WBC	Normal	660 (70.2)	280 (29.8)	Contrast		6.32	76.1
	Decreased	68 (73.1)	25 (26.9)	0.867 (0.537–1.399)	0.558	5.34	77.3
	Increased	138 (73.8)	49 (26.2)	0.837 (0.587–1.193)	0.325	3.64	83.1
NE	Normal	673 (69.5)	295 (30.5)	Contrast		6.03	75.1
	Decreased	62 (72.1)	24 (27.9)	0.883 (0.541–1.442)	0.619	10.09	89.9
	Increased	131 (78.9)	35 (21.1)	0.610 (0.410–0.907)	0.015	3.69	84.0
LC	Normal	797 (70.3)	337 (29.9)	Contrast		6.10	78.1
	Decreased	64 (82.1)	14 (17.9)	0.517 (0.286–0.935)	0.029	5.31	66.7
	Increased	5 (62.5)	3 (37.5)	1.419 (0.337–5.971)	0.633	/	100
CD4	Normal	259 (67.4)	125 (32.6)	Contrast		5.76	76.6
	Decreased	162 (74.0)	57 (26)	0.729 (0.504–1.055)	0.094	5.93	78.3
	Increased	320 (71.1)	130 (28.9)	0.842 (0.627–1.131)	0.253	6.13	78.0
CD8	Normal	230 (90.4)	115 (9.6)	Contrast		6.96	79.6
	Decreased	393 (87.3)	157 (12.7)	0.799 (0.598–1.068)	0.130	5.62	77.5
	Increased	118 (86.3)	40 (13.7)	0.678 (0.444–1.035)	0.072	5.46	74.2
CD4/CD8 ratio	Normal	363(89.8)	95 (10.2)	Contrast		7.80	79.3
	Decreased	75 (83.2)	17 (16.8)	0.496 (0.284–0.866)	0.014	4.86	71.4
	Increased	303 (87.2)	129 (12.8)	0.931 (0.706–1.227)	0.612	5.50	77.3
LC decline ratio	≥50%	6 (85.7)	1 (14.3)	1.35 (0.144–12.644)	0.793	5.14	66.7
	25–50%	18 (85.7)	3 (14.3)	1.35 (0.326–5.586)	0.679	5.24	66.7
	0–25%	40 (81.3)	9 (18.4)	Contrast		5.47	66.7

^1^ NLR: negative likelihood ratio. ^2^ NPV: negative predictive value.

**Table 5 tropicalmed-07-00278-t005:** Multivariate risk analysis of false-positive and true-negative QFT-GIT results in the diagnosis of active tuberculosis.

Influencing Factor	Group	True Negative n (%)	False Positive n (%)	OR (95% CI)	*p*
Age	<18 years	3 (100)	0 (0)		SE = 0
	18–40 years	97 (76.4)	30 (23.6)	0.650 (0.408–1.037)	0.071
	40–60 years	284 (73.6)	102 (26.4)	0.713 (0.529–0.960)	0.026
	≥60 years	357 (66.5)	180 (33.5)	Contrast	
Sex	Male	384 (64.5)	211 (35.5)	1.974 (1.484–2.625)	<0.001
	Female	357 (77.9)	101 (22.1)		
Diabetes	Yes	91 (61.5)	57 (38.5)	0.689 (0.459–1.033)	0.071
	No	857 (72.4)	327 (27.6)		
Immune system diseases	Yes	34 (91.9)	3 (8.1)	3.937 (1.171–13.239)	0.027
	No	914 (70.6)	381 (29.4)		
NE	Normal	586 (69.0)	263 (31.0)	Contrast	
	Decreased	56 (74.7)	19 (25.3)	0.848 (0.486–1.480)	0.563
	Increased	99 (76.7)	30 (23.3)	0.609 (0.389–0.952)	0.029
LC	Normal	797 (70.3)	337 (29.9)	Contrast	
	Decreased	64 (82.1)	14 (17.9)	0.537 (0.277–1.043)	0.066
	Increased	5 (62.5)	3 (37.5)	1.387 (0.303–6.357)	0.673
CD4/CD8 ratio	Decreased	64 (82.1)	14 (17.9)	Contrast	
	Increased	5 (62.5)	3 (37.5)	0.888 (0.667–1.183)	0.417
	Increased	303 (87.2)	129 (12.8)	0.527 (0.297–0.935)	0.028

**Table 6 tropicalmed-07-00278-t006:** Optimal cutoff values of QFT-GIT in the diagnosis of active tuberculosis in different influencing conditions.

Influencing Factor	Group	Cutoff Value	Sensitivity (%)	Specificity (%)
Age	<18 years	0.385	98.3	100
	18–40 years	0.655	87.4	85.4
	40–60 years	0.355	86.4	74.1
	≥60 years	0.485	79.13	71.4
Tumor	Yes	0.205	83.3	66.5
	No	0.655	82.4	78.0
Diabetes	Yes	1.035	81.6	76.4
	No	0.355/0.655	87.5/81.3	72.8/78.9
Immune system diseases	Yes	0.36	88.9	91.9
	No	0.655	82.0	77.5
WBC	Normal	0.655	83.4	77.7
	Decreased	0.315	88.3	73.4
	Increased	0.13	87.6	69.1
NE	Normal	0.655	82.7	76.8
	Decreased	0.425	92.7	73.6
	Increased	0.13	86.2	73.7
LC	Normal	0.655	83.2	77.0
	Decreased	0.275	86.0	80.8
	Increased	0.665	100	75.0

**Table 7 tropicalmed-07-00278-t007:** Comparison of the diagnostic value of QFT-GIT after adjusting cutoff values.

Influencing Factor	Cutoff Value	Sensitivity (%)	Specificity (%)	PPV ^1^ (%)	NPV ^2^ (%)
Ages (≥60 years)	0.35	82.1	67.7	68.2	81.8
	0.485	79.13	71.4	69.9	80.3
Tumor	0.35	76.7	69.1	43.8	90.4
	0.205	83.3	66.5	43.9	92.7
Diabetes	0.35	90.6	61.5	82.1	77.1
	1.035	81.6	76.4	87.0	68.1

^1^ PPV: positive predictive value. ^2^ NPV: negative predictive value.

## Data Availability

The data presented in this study are available on request from the corresponding author. The data are not publicly available due to patient’s privacy.

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
