# Peer review of "Factors Influencing False-Negative Results of QuantiFERON-TB Gold In-Tube (QFT-GIT) in Active Tuberculosis and the Desirability of Resetting Cutoffs for Different Populations: A Retrospective Study"

_tropicalmed, 2022, doi:10.3390/tropicalmed7100278_

Round 1

Reviewer 1 Report

The authors of the manuscript entitled "Factors influencing false-negative results of QTF-GIT in active tuberculosis and the desirability of resetting cutoffs for different populations: a retrospective study" estimated the false-negative and false-positive results of QuantiFERON Gold In-Tube (QTF-GIT) in the Chinese population. It was the retrospective study carried out with use the statistical methods.  The analyzed issue is important in term of the correct interpretation of test results when diagnosing tuberculosis.

In my opinion the manuscript may be accept for publication after following corrections:

1. In chapter Introduction  authors should describe more about Quantiferon type tests (especially principle and immunological basis of application) citing more new references.

2. In chapter Materials and Methods  in description of QTF-GIT the authors should complete the manufacturer's data of used Nil tubes, TB Ag tubes and Mitogen tubes.

3. In chapter Results under each table the authors should explain the letter abbreviations used in tables.

Reviewer 2 Report

The manuscript explores the factors influencing the false-negative results of QFT-GIT in active TB patients from China. It requires major language revision to enhance the understanding of the content.

1.     Line 34: Mycobacterium tuberculosis should be italic.

2.     Line 97-103: Please specify how many replicates were performed.

3.     Line 113 and 205: P (probability) should be in a lowercase letter. Please make it consistent throughout the manuscript.

4.     Line 128: Revise “exclued” to “excluded”.

5.     Line 212-256: The discussion requires major revision. I recommend comparing the results obtained with previous results and justify the findings.

6.     Line 299-377: Please revise the references following the journal’s requirements. 

Reviewer 3 Report

Yu et al has made an interesting and important study. However, it is crucial to underline that the use of QFT is not intended for diagnosing active TB alone. This need to be elaborated on, both in the introduction and in the discussion.

The discussion needs a big revision. It is unstructured and is an enumeration of other studies not put in context to the present study.

Minor comments:

QFT-GIT is not an abbreviation I have seen before. In the product insert they use QFT-Plus, it makes both the title and the abstract difficult to read, because it not explained until line 47.

Line 43: Bacterial negative TB, replace with culture  negative (throughout the manuscript)

Line 45: I suggest to include something about the difference/spectrum between latent TB, subclinical disease, etc.

Line 55-57: Interesting objective!

Line 102: Using the cut-off described by the manufacture?

Line 195: I suggest to include something about the few tests the cut-off value is based on.

Line 207: Could they be linked with LTBI?

Tables: Explain abbreviations PLR, PPV etc.

Table 2, footnote 3. Replace DR-TB with RR-TB

Round 2

Reviewer 2 Report

I recommend accepting after a language revision. There are many typos and grammatical errors. 

Author Response

Dear Reviewer:

Thank you again for your constructive comments concerning the manuscript entitled “Factors influencing false-negative results of QuantiFERON-TB Gold In-Tube (QFT-GIT) in active tuberculosis and the desirability of resetting cutoffs for different populations: A retrospective study (tropicalmed-1890191)”. Being a non-native English speaker, I am very sorry that this manuscript needs a lot of extensive English revision. We have sent the manuscript to professional English editing agency for further re-edting. Grammar and spelling errors have been corrected. The changes marked in right by using the track changes mode in MS Word. Here are some samples:

1. Line 44:Therefore, early diagnosis and treatment are effective strategies for prevention the further spread of MTB infection.

Change: Therefore, early diagnosis and treatment are effective strategies for preventing the further spread of MTB infection.

2. Line 66:However, this test cannot distinguish between active TB and LTBI.

Change: However, this test cannot be used to distinguish between active TB and LTBI.

3. Line 106: QFT-GIT was performed when a patient was suspected of TB. In this study,

Change: QFT-GIT was performed when a patient was suspected of having TB. 

4. Line123: The specific criteria are as following:

Change: The specific criteria are as follows:

5. Line 167: Others results were categorized as uncertain.

Change: Other results were categorized as uncertain .

6. Line176: To determine the comprehensive effect of influencing factors of true/false-negative and true/false-positive QFT-GIT results,

Change: To determine the comprehensive effect of influencing factors on true/false-negative and true/false-positive QFT-GIT results,

7. Line 255: and neutrophil counts still had a significant influence on false-positives QFT-GIT results

Change: and neutrophil counts still had a significant influence on false-positive  QFT-GIT results

8.Line276: We also selected several influencing factors to confirm the diagnostic value of the QFT-GIT after the adjustment of the cutoff values

Change: We also selected several influencing factors to confirm the diagnostic value of the QFT-GIT after adjusting the cutoff values

9. Line319: or from whom effective sputum cannot be obtained,

Change: or from whom effective sputum samples cannot be obtained

There are many changes in the manuscript to make the language more fluent.Thank you again for your positive comments and suggestions on our manuscript.

The English-Editing-Certificatethe was in the attachment.

All the best!

2022.09.21

--
Yours sincerely Dr Yuanyuan Yu
Department of Tuberculosis, Shanghai Pulmonary Hospital, Tongji University School of Medicine, Shanghai, China No. 507 Zhengmin
Road, Shanghai, China 200433 Shanghai China
Email: [email protected] 
